# Multiset Transformer:
# Advancing Representation Learning in Persistence Diagrams

## Abstract

To improve persistence diagram representation learning, we propose Multiset Transformer. This is the first neural network that utilizes attention mechanisms specifically designed for multisets as inputs and offers rigorous theoretical guarantees of permutation invariance. The architecture integrates multiset-enhanced attentions with a pool-decomposition scheme, allowing multiplicities to be preserved across equivariant layers. This capability enables full leverage of multiplicities while significantly reducing both computational and spatial complexity compared to the Set Transformer. Additionally, our method can greatly benefit from clustering as a preprocessing step to further minimize complexity, an advantage not possessed by the Set Transformer. Experimental results demonstrate that the Multiset Transformer outperforms existing neural network methods in the realm of persistence diagram representation learning.

## 1 Introduction

In recent years, the field of machine learning has seen the growing importance of multisets, often referred to as bags. These multisets provide an advanced form of traditional sets by allowing for multiple instances of identical elements. Descriptors based on histograms and the bag-of-words model, both examples of multisets, are not only common in Multiple Instance Learning (Dietterich et al., 1997; Babenko et al., 2010; Quellec et al., 2017), but also in natural language processing and text mining. Furthermore, they are recognized as among the primary representation techniques for tasks such as object categorization, as well as image and video recognition (Dalal & Triggs, 2005; Zhang et al., 2010; Welleck et al., 2018). This widespread use highlights the increasing importance and versatility of multisets in machine learning. Given their prevalence and critical role, understanding and mastering representation for multisets becomes essential.

In the realm of topological data analysis (TDA), persistence diagrams (PDs) stand out as a pivotal descriptor for persistent homology (PH), offering a multi-scale depiction of a space's intrinsic topological attributes (Edelsbrunner et al., 2002). Such topological insights have proven instrumental in deciphering complex biological structures, such as neural connections (Giusti et al., 2016), and have found applications in material science for analyzing porosity and nanomaterial structures (Nakamura et al., 2015). The recent fusion of PH-derived features with machine learning has enhanced both model performance and interpretability (Hofer et al., 2017; Horn et al., 2021). Nevertheless, the intrinsic multiset characteristics of PDs, as illustrated in Figure 1, present significant challenges for their direct integration into traditional machine learning models. These models predominantly necessitate inputs in a vector format, thus complicating the utilization of PDs. This essential transformation of PDs into vectors is known as vectorization (Ali et al., 2023), which align with our MST architecture.

Our contributions in this paper can be outlined in two primary areas: the introduction of a novel Multiset Transformer (MST) and its subsequent application in PD representation learning. The MST architecture is uniquely tailored to accept multisets as input, leveraging multiplicities to allocate greater attention to items with higher frequencies. This innovative design allows MST to utilize the multiplicities in multiset inputs while significantly reducing spatial and computational complexity compared to the Set Transformer. When applied to PD representation learning, MST consistently outperforms the existing methodologies across a

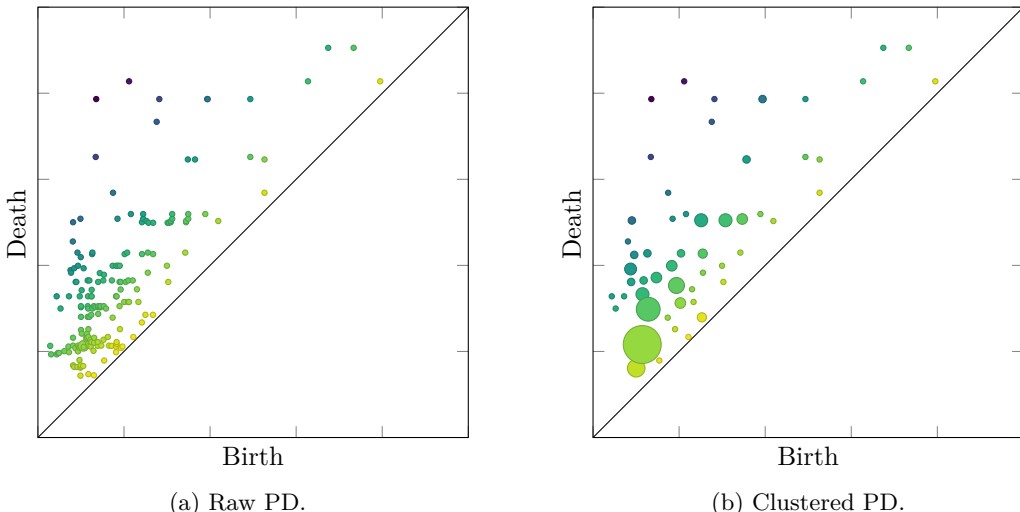

(a) Raw PD.

(b) Clustered PD.

Figure 1: Persistence diagram examples. PDs are represented as point sets in $\mathbb{R}^2$ above the diagonal. Each point's size indicates its multiplicity, and its color reflects the distance from the diagonal. The left PD contains 184 points, with 183 being distinct. After applying DBSCAN clustering, the diagram is reduced to 54 distinct points, as depicted in the right figure. Both PDs are characterized as multisets.

majority of the datasets we evaluated. Furthermore, our experimental results indicate that MST inherently offers an approximation by clustering the multiset prior to processing.

## 2 Related Works

The transformer model, as introduced by Vaswani et al. (2017), has revolutionized the field of deep learning, particularly within natural language processing (NLP). Its inception has spurred the development of numerous variants tailored to various data structures and applications, as outlined in the extensive surveys by Lin et al. (2022) and Khan et al. (2022). A prominent example is the Set Transformer (Lee et al., 2019), which offers a novel approach to processing unordered sets. However, to the best of our knowledge, this work presents the first transformer variant specifically designed for handling multisets.

Persistent homology and its PD vectorization have played a pivotal role in the integration of TDA into machine learning, as evidenced by works such as Dey & Wang (2022) and Ali et al. (2023). Traditional vectorization methods, including persistence landscapes (Bubenik et al., 2015) and persistence images (Adams et al., 2017), have garnered significant attention. The comprehensive survey by Ali et al. (2023) offers a structured framework, clarifying the various vectorization techniques available. In addition to these foundational methods, there has been a growing interest in adapting machine learning architectures to this domain. For instance, Hofer et al. (2017) introduced an innovative input layer for deep neural networks that processes topological signatures, computing a learnable parameterized projection. This idea was further refined by the same authors, who developed a neural network layer adept at handling barcodes by projecting points using parameterized functionals, as detailed in Hofer et al. (2019). Another notable contribution is PersLay, a specialized neural network layer for processing PDs, proposed by Carrière et al. (2020). The Persformer architecture, introduced by Reinauer et al. (2021), exemplifies the application of transformers to PD vectorization, presenting a transformer-centric methodology for PDs. Notably, our contribution diverges from the aforementioned works in a distinctive manner. The MST uniquely treats PDs as multisets rather than viewing them as lists of points that may be duplicated. This approach enables the model to incorporate a clustering phase before vectorization, effectively tackling the inherent computational challenges within this domain.

## 3 Preliminaries

### 3.1 Multisets and Permutation Properties

A multiset can be viewed as an extension of the conventional set, permitting the inclusion of multiple occurrences of identical elements. Formally, a multiset is defined by a *base set* $X$ accompanied by its multiplicity map $M : X \to \mathbb{Z}^+$. To illustrate, consider the multiset $\{a, b, b\}$. Here, the base set $X$ is represented as $\{a, b\}$, and the associated multiplicity map $M$ is such that $M(a) = 1$ and $M(b) = 2$.

Inherent to their definitions, both sets and multisets are indifferent to the order of their elements. Consequently, functions defined on sets or multisets inherently possess the property of permutation invariance.

**Definition 3.1** (Permutation Invariance). A function $f$ is said to be permutation invariant if the order of the components in its input vector does not influence its output. Formally, given any permutation $\sigma$ of the set of indices $\{1, 2, \ldots, n\}$ corresponding to a vector $\mathbf{x}$, $f$ is permutation invariant if and only if:

$$f(x_1, x_2, \ldots, x_n) = f\left(x_{\sigma(1)}, x_{\sigma(2)}, \ldots, x_{\sigma(n)}\right), \tag{1}$$

for all such permutations $\sigma$.

Conversely, while the concept of permutation invariance revolves around the indifference to order, there exists a contrasting property where the order is preserved, termed as permutation equivariance.

**Definition 3.2** (Permutation Equivariance). A vector-valued function $f = (f_1, \ldots, f_n)$ is characterized as permutation equivariant if any permutation applied to the input vector's elements results in a corresponding permutation of the output components, ensuring that the functional relationship remains intact. Formally, for every permutation $\sigma$ of the set of indices $\{1, 2, \ldots, n\}$:

$$f\left(x_{\sigma(1)}, \ldots, x_{\sigma(n)}\right) = \left(f_{\sigma(1)}(\mathbf{x}), \ldots, f_{\sigma(n)}(\mathbf{x})\right), \tag{2}$$

for all such permutations $\sigma$.

While Zhang et al. (2021) introduced the above permutation equivariance as set equivariance and further proposed multiset-equivariance as its relaxation, our architecture distinctly separates the base set from its multiplicities. This separation results in set like inputs, making set equivariance the appropriate property for our approach.

### 3.2 Pool-Decomposition Scheme

To achieve permutation invariance, the pool-decomposition scheme is commonly employed. Given a set $X = \{x_1, x_2, \ldots, x_n\} \in \mathcal{X}$, we define transformation functions $\phi : \mathcal{X} \to \mathbb{R}^h$ and $\rho : \mathbb{R}^h \to \mathbb{R}^d$, where $h$ and $d$ denote the hidden and output dimensions, respectively. The operator pool signifies a permutation-invariant pooling operation, such as sum, average, or max. Thus, a function $f$ on $X$ is expressed as:

$$f(X) = \rho\left(\text{pool}_{i=1}^n \phi(x_i)\right) \tag{3}$$

This approach is referenced in multiple studies, including Ravanbakhsh et al. (2016); Qi et al. (2017); Zaheer et al. (2017); Carrière et al. (2020); Reinauer et al. (2021). In our work, we adopt the pool-decomposition schema as a cornerstone for our MST design.

## 4 Problem Setup

From a machine learning perspective, the problem can be formally described as follows: Let $\mathcal{D}$ denote a bounded, finite multiset space representing the data space, let $\mathcal{H}$ denote a Hilbert space serving as the feature space, and let $\mathcal{Y}$ represent the target space. Consider a learning system characterized by the mappings

$$\mathcal{D} \xrightarrow{f_\theta} \mathcal{H} \xrightarrow{g_\phi} \mathcal{Y},$$

where the function $f_\theta$, parameterized by $\theta$, maps elements of $\mathcal{D}$ to representations in $\mathcal{H}$, and the function $g_\phi$, parameterized by $\phi$, predicts outcomes in $\mathcal{Y}$ based on these representations. The objective is to develop and refine the representation $f_\theta$ such that $g_\phi$ can achieve better predictions by minimizing the loss function $L(g_\phi(f_\theta(x)), y)$, where $(x, y) \in \mathcal{D} \times \mathcal{Y}$. To illustrate, in this paper, we employ MST as the model for the representation function $f_\theta$.

From an application perspective, the problem centers on acquiring representations of PDs suitable for classification tasks. These PDs are derived from graphs, which are fundamentally multisets, as depicted in Figure 1. Inherently, PDs do not possess intrinsic bounds or finite limits; yet, for practical purposes, they are typically preprocessed into bounded, finite multisets. For the purposes of this discussion, it is presupposed that PDs are treated as bounded and finite multisets.

This representation learning framework adheres to key criteria outlined subsequently:

1. **Permutation Invariance**: Given the fundamental properties of multisets, the representation must be order-agnostic. This ensures a consistent representation, irrespective of the ordering of elements.

2. **Practical Feasibility**: The training process for representations should be computationally feasible, emphasizing feasibility and scalability, particularly for large-scale datasets.

3. **Feature Retention**: It is imperative for the representation to capture and retain key features. This enables subsequent machine learning tasks to effectively exploit the data's underlying structure.

By addressing these priorities, we aim to bridge the gap between multiset representation learning and its applications ensuring that the complex structure of the data remains most informative throughout the representation process.

## 5 Multiset Transformer

The Multiset Transformer (MST) is based on the fundamental idea of leveraging the multiplicity inherent in multisets to influence the attention score function. Essentially, the objective is to steer the model's attention towards items with relatively higher multiplicities. This is grounded in the belief that items with higher multiplicities may carry more importance or relevance in certain contexts, and thus should be given more weight during the attention process. To provide a comprehensive explanation of the MST, it is imperative to first explain the underlying mechanisms of attention.

### 5.1 Scaled Dot-Product Attention Mechanism

The attention mechanism we employ is consistent with the one presented by Vaswani et al. (2017). Given $n$ query vectors $Q \in \mathbb{R}^{n \times d_q}$, $m$ key vectors $K \in \mathbb{R}^{m \times d_k}$, and $m$ value vectors $V \in \mathbb{R}^{m \times d_v}$, where $d_q$, $d_k$, and $d_v$ represent their respective dimensions, the attention function $\text{Att}(Q, K, V)$ is designed to map queries $Q$ to outputs using the key-value pairs. This is mathematically represented as:

$$\text{Att}(Q, K, V; \omega) = \omega(QK^\top)V. \tag{4}$$

The pairwise dot product $QK^\top \in \mathbb{R}^{n \times m}$ serves as a measure of similarity between each pair of query and key vectors. The weights for this dot product are computed using the activation function $\omega$. Consequently, the output $\omega(QK^\top)V$ is essentially a weighted sum of $V$. Here, a value vector from $V$ receives a higher weight if its associated key vector has a larger dot product with the query. Note that the dot product $QK^\top$ necessitates $d_q = d_k$ for dimensional consistency.

Building upon this, we specifically utilize the scaled dot-product attention in our construction, which is given by:

$$\text{Att}(Q, K, V) = \text{softmax}\left(\frac{QK^\top}{\sqrt{d_k}}\right)V. \tag{5}$$

This formulation ensures that the attention weights are appropriately normalized, and the scaling factor $\sqrt{d_k}$ aids in stabilizing the magnitudes of the dot products, especially when $d_k$ is large. Furthermore,

our architecture incorporates multihead attention, a technique that facilitates capturing a diverse range of relationships and dependencies within the input data. Each attention head operates independently, allowing the model to focus on different aspects of the input simultaneously. Additional details on multihead attention can be found in the Appendix for clarity.

## 5.2 Multiset Attention Mechanisms in the Transformer

Within the framework of the MST, attention is employed in two distinct manners: self-attention and attention with learnable queries. To clarify these mechanisms, we first establish the necessary notations and their interpretations.

A multiset is denoted as $(X, M_X)$, where $X$, the base set, is represented as an array of points in $\mathbb{R}^{n \times d}$. The multiplicities associated with these points are specified by $M_X \in \mathbb{R}^{n \times 1}$, where each element of $M_X$ corresponds to the multiplicity of the respective element in $X$. It is important to note that a multiset reduces to a conventional set when all elements of $M_X$ are equal to unity, that is, $M_i = 1$ for each $i$.

### 5.2.1 Multiset-Enhanced Attention Mechanism

In this section, we introduce the concept of multiset attention, a novel approach designed to handle input multisets and their associated multiplicities. The primary motivation behind this mechanism is to incorporate the multiplicity information into the traditional attention mechanism, thereby enhancing its ability to focus on elements with larger multiplicities.

Given input base sets $Q \in \mathbb{R}^{n \times d}$ and $X \in \mathbb{R}^{m \times d}$, along with their respective multiplicities $M_Q \in \mathbb{R}^{n \times 1}$ and $M_X \in \mathbb{R}^{m \times 1}$, we define the multiset-enhanced attention weights as:

$$A(Q, X) := \left( \text{softmax} \left( \frac{QX^\top}{\sqrt{d}} \right) + \alpha B \right) X, \tag{6}$$

$$B := \frac{(M_Q - \mathbf{1})(M_X - \mathbf{1})^\top}{\|(M_Q - \mathbf{1})(M_X - \mathbf{1})^\top\|_F + \varepsilon}. \tag{7}$$

Here, $\|\|_F$ denotes the Frobenius norm, which is utilized to normalize this new term. The parameter $\alpha$ is learnable, allowing the model to adjust its influence during training. $\varepsilon$ is used as a small constant to avoid zero division.

Intuitively, the introduction of the multiplicities term (7) enables the attention mechanism to allocate greater emphasis to elements with higher multiplicities. This enhancement augments the mechanism's ability to selectively focus on the most salient elements within a multiset.

### 5.2.2 Multiset Self-Attention

Self-attention, as described by Vaswani et al. (2017), is characterized by the attention scores being derived directly from the input. This mechanism inherently captures the intra-correlation present within the input data. Building upon the multiset attention framework discussed in the previous sections, we extend this concept to introduce the multiset self-attention.

Given an input matrix $X \in \mathbb{R}^{m \times d}$ with multiplicity $M_X \in \mathbb{R}^{m \times 1}$, multiset self-attention is achieved by setting $Q = K = V = X$, leading to the self-attention $A(X, X)$. This self-attention $A(X, X)$ is notable for its property of permutation equivariance, which we formalize in the following theorem.

**Theorem 5.1.** *The multiset self-attention, represented as $A(X, X)$, is permutation equivariant.*

### 5.2.3 Multset Attention with Learnable Queries

In the previous section, we introduce a permutation equivariant multiset self-attention. However, to ensure the permutation invariant property in a MST, it is necessary to incorporate a permutation invariant operator, i.e., the pool operator in Equation (3). The core of our approach lies in the multiset attention with learnable

queries. Specifically, we set $K = V = X$ and introduce a learnable matrix $Q \in \mathbb{R}^{n \times d}$, where $n$ is a user-defined parameter. It is worth noting that the query is independent of the input and is shared across all input instances, enabling the capture of common features during training. To account for multiplicity, we define the multiset attention with learnable queries as:

$$A_Q(X) := \left( \mathrm{softmax} \left( \frac{QX^\top}{\sqrt{d}} \right) + B \right) X, \tag{8}$$

$$B := \frac{M_\alpha (M_X - \mathbf{1})^\top}{\|M_\alpha (M_X - \mathbf{1})^\top\|_F + \varepsilon}. \tag{9}$$

where $M_\alpha \in \mathbb{R}^{n \times 1}$ are learnable parameters. Similarly, we formalize its permutation invariant in the following theorem.

**Theorem 5.2.** *The multiset attention with learnable queries $A_Q(X)$ is permutation invariant.*

The terms introduced in Equations (7) and (9) are not the only possible choices, yet they are carefully selected for their alignment with several key aspects of our framework. Firstly, these multiplicity bias terms successfully incorporate biases associated with multiplicities into the attention weights, aligning with our philosophy of assigning attention based on multiplicities. Moreover, they are key in maintaining the permutation equivariant or invariant properties, which are important for our model's integrity. Finally, they demonstrate mathematical consistency by reducing to zero for set inputs, when $M_Q = \mathbf{1}$ and $M_X = \mathbf{1}$, these attentions degenerate to the original ones.

### 5.3 Multiset Attention Blocks

Multiset self-attention and multiset attention with learnable queries are crucial in the construction of the MST. To ensure clarity and maintain consistency with existing literature, we adopt and slightly modify terminologies introduced by Lee et al. (2019).

The cornerstone of our proposed architecture is the Multiset Attention Block (MAB). This block can be described using the following formulation:

$$\mathrm{MAB}(Q, X) := \mathrm{LN}(H + \mathrm{FFN}(H)), \tag{10}$$

$$H := \mathrm{LN}(Q + A(Q, X)). \tag{11}$$

Here, FFN denotes a position-wise feedforward layer, specifically row-wise. The symbol LN represents layer normalization, as illustrated by Ba et al. (2016).

Based on the MAB, we construct several foundational components important for the MST. The multiset attention block with learnable queries, denoted as $\mathrm{MAB}_Q$, is formulated for a given set $X$. Mathematically, it can be expressed as:

$$\mathrm{MAB}_Q(X) := \mathrm{LN}(H + \mathrm{FFN}(H)), \tag{12}$$

$$H := \mathrm{LN}(Q + A_Q(X)). \tag{13}$$

The Multiset Self-Attention Block (SAB) is a specialized case of the MAB wherein the input set serves as both the key and value. This block can be represented as:

$$\mathrm{SAB}(X) := \mathrm{MAB}(X, X). \tag{14}$$

Inspired by the Induced Set Attention Block (ISAB) as presented in Lee et al. (2019), we also introduce the Induced Multiset Attention Block (IMAB). This block is designed to provide similar functionality but with reduced computational demands. It is defined as:

$$\mathrm{IMAB}(X) := \mathrm{MAB}\left(X, \mathrm{MAB}_Q(X)\right). \tag{15}$$

It is evident that both SAB and IMAB maintain permutation equivariance. In contrast, $\mathrm{MAB}_Q$ exhibits permutation invariance.

## 5.4 Multiset Transformer Overall Architecture

The MST is formulated on the foundation of a pool-decomposition scheme, as articulated in Equation (3). The rationale for this architectural decision is bifurcated. Initially, it guarantees that the model attains an optimal level of expressiveness, facilitating the capture of complex patterns and interrelationships within the dataset. Concomitantly, it is imperative to note that the model embodies a permutation-invariant property, a requisite dictated by the inherent characteristics of a multiset. The overall architecture of the MST, detailing its various components and their interconnections, is illustrated in Figure 2.

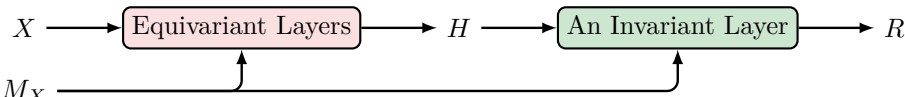

Figure 2: MST architecture. Base set $X$ with multiplicities $M_X$ is processed by equivariant layers, preserving permutation order. Representation output $R$ is generated, with multiplicities $M_X$ used as input to an invariant layer.

Within the given figure, a multiset is denoted as $(X, M_X)$, where $X$ represents the base set, and $M_X$ indicates its corresponding multiplicities. The architecture consists of two main components: the Equivariant Layers and an Invariant Layer. The Equivariant Layers are structured as a sequence of permutation equivariance blocks, which can be either SABs or IMABs. The outputs from these layers, denoted as $H$, possess permutation equivariance. Given that $H$ maintains permutation equivariance in relation to $X$, it follows that the multiplicities $M_X$ are intrinsically the multiplicities of $H$. This architectural design ensures that the multiplicities not only enrich the Equivariant Layers but also strengthen the subsequent Invariant Layer.

The Invariant Layer can be constructed either by employing invariant operators or by applying the $\text{MAB}_Q$ operation. This procedure ultimately results in the final output $R$, serving as the representation of the multiset. Therefore, a conventional architecture wherein the Equivariant Layers are constructed via IMAB and the Invariant Layer is performed by $\text{MAB}_Q$ can be expressed as:

$$H = \text{IMAB}(\text{IMAB}(\ldots \text{IMAB}(X)\ldots)), \tag{16}$$

$$R = \text{MAB}_Q(H). \tag{17}$$

This architecture ensures a balance between expressiveness and complexity, making it suitable for a wide range of applications.

## 5.5 Complexity Analysis of Multiset Transformer and Set Transformer

To thoroughly highlight the advantages of the MST, we conduct a complexity analysis comparing it to the Set Transformer (ST), particularly focusing on scenarios involving multiset inputs. Notably, the attention mechanism significantly influences the complexity of these architectures, forming the core of our analysis.

For a multiset $X \in \mathbb{R}^{n \times d}$, where $d$ is fixed, and accompanied by its multiplicity information $M_X \in \mathbb{R}^{n \times 1}$, it's important to acknowledge that the ST processes inputs in the form of lists. In cases where $M_X$ results in a multiset list, the size of the list can scale to $\mathcal{O}(nm)$, with $m$ representing the maximum multiplicity within $M_X$. Consequently, the complexities of the Set Attention Block (SAB) and Induced Set Attention Block (ISAB) operations in the Set Transformer are $\mathcal{O}(n^2 m^2)$ and $\mathcal{O}(nmq)$, respectively. Here, $q$ denotes the number of inducing points in ISAB.

In contrast, the MST, designed to handle multiplicities without duplicating elements, demonstrates complexities of $\mathcal{O}(n^2)$ and $\mathcal{O}(nq)$ for the SAB and IMAB, respectively. This comparison underscores the inherent lower complexity of the MST when processing multiset inputs. Similarly, the space complexity exhibits the same favorable results, with $d$ being held constant.

# 6 Experiments

In the experiments section, our studies are divided into two primary parts: synthetic experiments and persistence diagram (PD) representation learning. The goal of the synthetic experiments is to provide a preliminary study to confirm that the framework operates as expected. On the other hand, the PD representation learning aims to demonstrate the effectiveness and relevance of our framework on real-world datasets. Through these experiments, we attempt to substantiate both the theoretical foundations and the practical usefulness of our proposed approach.

## 6.1 Synthetic Experiments

In the synthetic experiments, our goal was to demonstrate the ability of the MST to highlight elements that appear with the highest frequency within a multiset. We employed synthetic samples composed of multisets of 2-dimensional points. The true label is determined based on the element that appears with the highest frequency. For the purpose of learning representations of these multisets, we employed the MST, both with and without multiplicity inputs. Subsequently, these representations were fed into a fully connected layer for prediction, as outlined in Figure 3.

$$(X, M_X) \longrightarrow \boxed{\text{MST}} \longrightarrow R \longrightarrow \boxed{\text{FC}} \longrightarrow Y$$

Figure 3: Sythetic data classification pipeline. A multiset sample $(X, M_X)$ is processed by a Multiset Transformer (MST) to generate its representation $R$. This representation is then used by a fully connected (FC) classifier to make predictions $Y$. In the context of problem setup (Section 4), MST corresponds to the representation function $f_\theta$ and FC corresponds to the task-specific function $g_\phi$.

With the given architecture, we utilized a ten-fold cross-validation method, which was conducted over 10 separate iterations. To elaborate, the entire dataset was partitioned into 10 equal folds. For each of these iterations, our model was trained using nine of these folds and subsequently tested on the one remaining fold. This ensured that every individual fold had a turn as the test set. The accuracy was averaged over these 10 iterations to yield the result for a single run. To enhance the robustness of our results, this entire process was repeated for a total of 5 runs, with each run shuffling the data randomly. The cumulative results, which include both the mean and standard deviation from all 5 runs, are presented in Table 1.

Table 1: Results on synthetic datasets. In this table, '# Classes' refers to the total number of class labels in the dataset. 'Ratio' is defined as the data ratio, calculated using $\frac{|X|}{\|M_X\|_1}$. 'MST (w/o mult.)' represents the Multiset Transformer without multiplicity inputs. 'MST' denotes the Multiset Transformer. Note that all ratios are presented in decimal form, and prediction accuracies are expressed as percentages.

| # Classes | Ratio | MST (w/o mult.) | MST |
|---|---|---|---|
| 2 | 0.03 | $55.94_{\pm 0.50}$ | $\mathbf{100.00}_{\pm 0.00}$ |
| 3 | 0.03 | $39.92_{\pm 0.58}$ | $\mathbf{99.88}_{\pm 0.15}$ |
| 5 | 0.04 | $26.72_{\pm 0.72}$ | $\mathbf{88.86}_{\pm 2.07}$ |
| 11 | 0.05 | $15.06_{\pm 0.24}$ | $\mathbf{41.14}_{\pm 2.24}$ |

To ensure consistency in our experiments, hyperparameters were kept constant across all configurations. These hyperparameters are detailed in the Appendix. In the table, the data ratio is defined as $\frac{|X|}{\|M_X\|_1}$, where $|X|$ is the cardinality of the base set $X$. It represents the ratio of the count of unique items to the sum of their multiplicities, highlighting the size difference between set and multiset representations.

Table 1 demonstrates the enhanced performance of the MST when equipped with multiplicity. Specifically, this advantage is clearly shown in scenarios involving 2 or 3 classes, where MST achieves near-perfect or perfect accuracies, significantly surpassing its counterpart. Intriguingly, the relative improvements for datasets with 5 or 11 classes, measured at approximately 232% and 173%, respectively, are higher than

those observed in the 2 and 3 class cases (79% and 150%). This suggests that with an increasing number of classes, MST not only maintains its lead but does so with a more pronounced relative margin, despite a seemingly narrower absolute margin of improvement. These findings highlight the successful achievement of MST's design objectives, validating its effectiveness in leveraging the attributes of multiplicity in multisets.

## 6.2 Persistence Diagram Representation Learning

In this segment of our study, we concentrate on validating the efficacy of the MST as *a neural network vectorization method for PDs*. To our knowledge, PERSLAY (Carrière et al., 2020) currently maintains state-of-the-art (SOTA) benchmarks in the majority of its datasets within this specialized field. Consequently, we designate PERSLAY as our comparative baseline.

### 6.2.1 Dataset Selection and Evaluation Metrics

Adapting the approach proposed by Carrière et al. (2020), this study harnesses PD representation learning for graph datasets. Specifically, our focus gravitates towards chemoinformatics network datasets such as MUTAG, COX2, DHFR, NCI1, and NCI109 (Debnath et al., 1991; Sutherland et al., 2003; Wale et al., 2008; Shervashidze et al., 2011). Beyond this, we delve into the bioinformatics dataset PROTEIN (Dobson & Doig, 2003) and larger real-world social networks, which include movie (IMDB-BINARY, IMDB-MULTI) and scientific collaboration networks (COLLAB) (Yanardag & Vishwanathan, 2015). As classification is the primary goal for all datasets in focus, we utilize classification accuracy as the evaluation criterion. An extensive analysis and quantitative overview of these datasets can be found in the Appendix.

### 6.2.2 Persistence Diagrams

In our experimental framework, we primarily utilize two types of topological features: ordinary PDs and extended PDs. These diagrams are represented as multisets in $\mathbb{R}^2$, denoted as $X_1, X_2, X_3, \ldots$, where the subscripts indicate different types of PDs resulting from various filtrations. When combined with their corresponding multiplicities, we have pairs of PDs and their multiplicities, such as $(X_1, M_{X_1}), (X_2, M_{X_2}), (X_3, M_{X_3}), \ldots$.

The computation of persistence values relies on filtrations, which are in turn generated by specific functions. In the context of this research, we primarily employ heat kernel signatures (HKS) as these specific functions to maintain consistency with the settings in Carrière et al. (2020). A more comprehensive explanation of these features, along with the selection of filtration methods, is provided in the Appendix for further clarity.

### 6.2.3 Graph Classification Architecture

To ensure a rigorous and unbiased comparison, we adopt the experimental settings described in Carrière et al. (2020). Our network comprises two main components: first, we employ MSTs to effectively learn representations of PDs, and subsequently, we use a fully connected layer for prediction tasks. A visual representation of this architecture can be found in Figure 4.

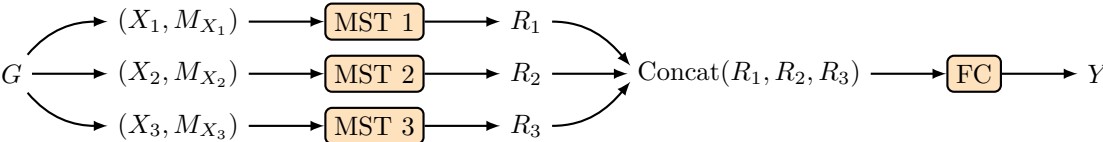

Figure 4: Graph classification architecture. Given a graph $G$, it's encoded into various ordinary or extended PDs, i.e., $(X_i, M_{X_i})$. Here, we have $i \in \{1, 2, 3\}$ as demonstration. In experimentation, we can have $i \in \{1, 2, \ldots, n\}$ for some finite integer $n$. Each diagram is processed by an independent instance of the Multiset Transformer (MST $i$), yielding its representation $R_i$. These representations are concatenated to form the complete feature set of the graph. The classifier, represented by a fully connected layer (FC), then makes predictions based on this comprehensive representation.

### 6.2.4 Main Results

Similar to the experiments in Subsection 6.1, we applied a ten-fold cross-validation over 10 iterations on our dataset. Each iteration trained the model on nine folds and tested on one, ensuring all folds served as the test set. After averaging the accuracy over the 10 iterations for a single run, we repeated this process for 5 runs with random shuffling of data each time. The aggregated results, including mean and standard deviation from the 5 runs, are detailed in Table 2.

Table 2: PD representation learning results. This table utilizes the abbreviation 'PSL' to denote PERSLAY, 'R' to signify the data ratio, which is computed as $\frac{|X|}{\|M_X\|_1}$, and 'MST' to represent the Multiset Transformer. Data ratios are displayed in decimal form, whereas all accuracy measures are provided in percentage format. All the results of PERSLAY are copied from Carrière et al. (2020, Table 7).

| Datasets | Ordinary PDs | | | | | Extended PDs | | | | |
| | PSL | w/o Clustering | | w/ Clustering | | PSL | w/o Clustering | | w/ Clustering | |
| | | R | MST | R | MST | | R | MST | R | MST |
|---|---|---|---|---|---|---|---|---|---|---|
| MUTAG | 70.2 | 0.91 | **89.25**±0.77 | 0.14 | **87.03**±1.13 | 85.1 | 1.00 | **89.54**±1.32 | 0.51 | **90.50**±0.67 |
| COX2 | 79.0 | 0.92 | **81.20**±1.04 | 0.04 | **80.67**±0.27 | 81.5 | 1.00 | **81.61**±0.51 | 0.47 | 79.96±1.18 |
| DHFR | 71.8 | 0.90 | **74.12**±0.37 | 0.04 | **76.37**±0.18 | **78.2** | 0.99 | 73.18±0.49 | 0.48 | 71.55±0.70 |
| NCI1 | 68.9 | 0.98 | **69.12**±0.17 | 0.22 | 68.65±1.23 | **72.3** | 0.99 | 70.28±0.12 | 0.69 | 69.57±0.19 |
| NCI109 | 66.2 | 0.97 | **66.38**±0.40 | 0.22 | 64.87±0.51 | 67.0 | 0.99 | **68.75**±0.27 | 0.68 | **67.11**±0.45 |
| PROTEIN | 69.7 | 0.99 | **72.92**±0.47 | 0.37 | **72.89**±0.26 | 72.2 | 0.94 | **75.49**±0.44 | 0.21 | **74.16**±0.22 |
| IMDB-B | 64.7 | 0.99 | **70.76**±0.47 | 0.70 | **70.66**±0.42 | 68.8 | 0.49 | **75.40**±0.14 | 0.06 | **74.72**±0.42 |
| IMDB-M | 42.0 | 0.99 | **45.44**±0.35 | 0.78 | **44.95**±0.48 | 48.2 | 0.44 | **51.46**±0.29 | 0.06 | **50.33**±0.17 |
| COLLAB | 69.2 | 0.99 | **69.90**±0.32 | 0.61 | **70.42**±0.26 | 71.6 | 0.52 | **74.18**±0.38 | 0.01 | **72.26**±0.21 |

In our results, we primarily showcase two variations of the model: the MST applied directly and the MST preceded by a clustering preprocess. Detailed comparisons and visualizations of the MST model with the clustering preprocess can be found in Figure 5. In our experiments, we utilize the DBSCAN clustering algorithm, with the hyperparameters detailed in the Appendix. Our primary goal was to assess the effectiveness of the MST in the context of PD representation learning, and the results yielded valuable insights on multiple fronts.

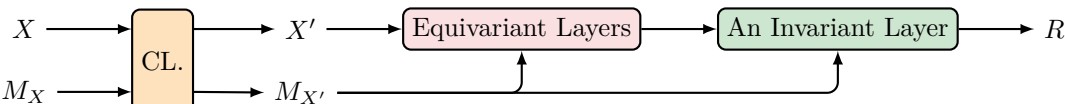

Figure 5: Multiset Transformer architecture with clustering preprocessing. In this architecture, 'CL.' represents the clustering preprocessing step. Initially, the input multiset $(X, M_X)$ undergoes clustering to approximate its structure. The output of this preprocessing stage is a clustered multiset, denoted by $(X', M_{X'})$, which then serves as the input for the Multiset Transformer.

Firstly, the results demonstrate that, across a majority of datasets, the performance of the MST without clustering preprocessing exceeds that of PERSLAY. This trend holds true for both Extended and Ordinary PDs, thereby highlighting the robustness and superiority of the MST in the task of PD vectorization using neural networks.

In comparing the performance of the MST with and without clustering, it may initially be hypothesized that the incorporation of clustering could detrimentally impact model performance due to data simplification. Contrary to this intuition, our empirical results reveal that the integration of clustering with the MST generally results in only a marginal decrease in accuracy. This is particularly notable in the case of Extended PDs for the COLLAB dataset, where even when reduced to just 0.01 of the original input size, the performance still surpasses that of the baseline model. Given the substantial benefits in terms of computational complexity reduction and improved scalability, this slight trade-off in accuracy is deemed justifiable.

More interestingly, in specific scenarios such as Extended PDs for the MUTAG and Ordinary PDs for DHFR and COLLAB, clustering not only mitigates the anticipated performance degradation but also actively boosts accuracy. This nuanced observation suggests that clustering might capture intrinsic data structures beneficial for representation in certain contexts.

### 6.2.5 Ablation Study

In this section, we conduct an ablation study to examine the entries presented in Table 2. The primary aim of this analysis is to demonstrate the performance improvements attributable to the introduction of multiplicity terms. To thoroughly investigate the influence of multiplicities on our model, we specifically focus on entries characterized by lower data ratios in Table 2. We present the results in Table 3.

Table 3: Ablation analysis of PD representation learning. This table employs the abbreviation 'PD' to indicate the type of Persistence Diagram, either Ordinary or Extended, 'R' to denote the Data Ratio, which is expressed in decimal form, and 'MST' to represent the Multiset Transformer. The results of column MST are the same as those in Table 2. The variants of MST are described as 'MST (w/o mult.)', indicating the Multiset Transformer excluding multiplicities, and 'MST (w/o PD)', denoting the Multiset Transformer excluding persistence diagrams. Accuracy measures are displayed in percentage format.

| Dataset | PD | R | MST | MST (w/o mult.) | MST (w/o PD) |
|---|---|---|---|---|---|
| MUTAG | Ord. | 0.14 | $\mathbf{87.03}_{\pm 1.13}$ | $85.63_{\pm 1.16}$ | $80.96_{\pm 0.78}$ |
| COX2 | Ord. | 0.04 | $\mathbf{80.67}_{\pm 0.27}$ | $80.63_{\pm 0.37}$ | $73.92_{\pm 0.65}$ |
| DHFR | Ord. | 0.04 | $\mathbf{76.37}_{\pm 0.18}$ | $76.19_{\pm 0.24}$ | $66.11_{\pm 0.56}$ |
| NCI1 | Ord. | 0.22 | $\mathbf{68.65}_{\pm 0.14}$ | $67.28_{\pm 0.18}$ | $57.11_{\pm 0.33}$ |
| NCI109 | Ord. | 0.22 | $\mathbf{64.87}_{\pm 0.51}$ | $63.35_{\pm 0.58}$ | $56.06_{\pm 0.16}$ |
| PROTEIN | Ext. | 0.21 | $\mathbf{74.16}_{\pm 0.22}$ | $72.76_{\pm 0.35}$ | $63.74_{\pm 0.46}$ |
| IMDB-B | Ext. | 0.06 | $\mathbf{74.72}_{\pm 0.42}$ | $71.10_{\pm 0.63}$ | $62.72_{\pm 0.75}$ |
| IMDB-M | Ext. | 0.06 | $50.33_{\pm 0.17}$ | $\mathbf{50.64}_{\pm 0.41}$ | $43.24_{\pm 0.39}$ |
| COLLAB | Ext. | 0.01 | $\mathbf{72.26}_{\pm 0.21}$ | $71.58_{\pm 0.26}$ | $40.30_{\pm 0.34}$ |

Consistently across all datasets, the MST model, when fully equipped with PD and its associated multiplicities, demonstrates superior performance over its counterparts. A direct comparison between the complete MST model and its variant, MST (w/o mult.), provides insights into the pivotal role of the multiplicities terms within the MST framework. Notably, the exclusion of these terms from the model's architecture invariably leads to a decrease in accuracy for most datasets. To illustrate, datasets such as MUTAG, NCI1, NCI109, PROTEIN, and IMDB-B exhibit an accuracy reduction exceeding 1%.

As anticipated, the omission of the entire PD results in a more pronounced decline in accuracy than merely excluding its multiplicities. Nevertheless, the recurrent performance dip across datasets, when solely removing multiplicities, accentuates their integral role in PD representation learning. For instance, in the IMDB-B dataset, the performance disparity between the full MST and MST (w/o mult.) stands at 3.62%, underscoring the multiplicities' influence on the efficacy of PD.

To summarize, this ablation analysis emphasizes the criticality of the multiplicities bias within the MST framework. Their consistent and positive impact on the MST's performance across various datasets is a testament to their importance. The enhanced performance due to multiplicities underscores the MST's robustness, as evidenced not only through synthetic experiments but also in real-world applications.

### 6.3 Limitations and Further Works

While the MST effectively meets our objectives by exhibiting strong performance on both synthetic and real-world datasets, there are certain limitations to our study that warrant attention.

Firstly, the improved accuracies achieved through clustering warrant a deeper exploration. A thorough investigation is essential to determine the underlying mechanisms contributing to this improvement.

Furthermore, it is widely acknowledged that the significance of a feature, specifically a point in the PD, is determined by its lifespan, which is schematically represented as the distance to the diagonal in Figure 1. The greater the lifespan, the more significant the feature. In our experiments, we did not employ this assumption. However, we suggest that this property could be more comprehensively integrated into representation learning. Such integration may pave the way for the development of more refined and effective models in the future.

## 7   Conclusion

In this paper, we present the Multiset Transformer, the first neural network based on attention mechanisms designed specifically for multisets as inputs, and it comes with rigorous theoretical guarantees of permutation invariance. This model leverages the multiplicities in a multiset by allocating more attention to elements with larger multiplicities. The synthetic experiments demonstrate that our model achieves the intended design goals. In the domain of persistent diagram vectorization using neural networks, our approach surpasses existing state-of-the-art methods across most datasets. Furthermore, with its inherent ability to approximate sets into multisets via clustering, the Multiset Transformer offers a viable solution for datasets of *any size*, provided they can be efficiently clustered.

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

# A   Comparative Analysis of Multiset Transformer and Set Transformer

In this section, we present a detailed performance comparison between the MST and the Set Transformer (ST), with the results detailed in Table 4. It is important to clarify that the MST column in Table 1 corresponds to the column labeled MST (inv.).

Table 4: Comprehensive results of synthetic datasets. In this table, '# Classes' denotes the total number of class labels in the dataset. 'Ratio' is defined as the data ratio, calculated using $\frac{|X|}{\|M_X\|_1}$. 'MST (w/o mult.)' signifies the MST excluding multiplicity inputs. Additional variants include 'MST (equiv. & inv.)', with multiplicities in both equivariant and invariant layers, and 'MST (inv.)', with multiplicities only in the invariant layer. 'ST' refers to the Set Transformer applied to full multisets. All ratios are presented in decimal form, and prediction accuracies are expressed as percentages.

| # Classes | Ratio | Random | MST (w/o mult.) | MST (equiv. & inv.) | MST (inv.) | ST |
|---|---|---|---|---|---|---|
| 2 | 0.03 | 50.00 | $55.94_{\pm 0.50}$ | $\mathbf{100.00}_{\pm 0.00}$ | $\mathbf{100.00}_{\pm 0.00}$ | $\mathbf{100.00}_{\pm 0.00}$ |
| 3 | 0.03 | 33.33 | $39.92_{\pm 0.58}$ | $98.74_{\pm 0.36}$ | $\mathbf{99.88}_{\pm 0.15}$ | $99.74_{\pm 0.15}$ |
| 5 | 0.04 | 20.00 | $26.72_{\pm 0.72}$ | $74.28_{\pm 1.19}$ | $\mathbf{88.86}_{\pm 2.07}$ | $85.82_{\pm 1.73}$ |
| 11 | 0.05 | 9.09 | $15.06_{\pm 0.24}$ | $34.60_{\pm 2.07}$ | $41.14_{\pm 2.24}$ | $\mathbf{42.02}_{\pm 1.85}$ |

There are two variants of MST: MST (equiv. & inv.) and MST (inv.). The primary distinction between these variants lies in whether multiplicities are incorporated into the equivariant layers. Specifically, MST (equiv. & inv.) takes into account the multiplicities, whereas MST (inv.) does not.

In contrast, the input for ST consists of the complete enumeration of elements in a multiset, including repeated elements. For instance, consider a multiset $\{a, b\}$ with multiplicities $M(a) = 2$ and $M(b) = 4$. In this case, the input for ST would be a sequence $\{a, a, b, b, b, b\}$, reflecting each instance of the elements in the multiset. This difference in handling input data between MST and ST is a crucial factor in their performance comparison.

Results from Table 4 indicate that the Set Transformer demonstrates comparable, and in certain instances, superior performance compared to various configurations of the MST. Notably, in the context of 11 classes, the ST marginally outperforms the MST, achieving a prediction accuracy of $42.02\% \pm 1.85\%$ as opposed to MST's $41.14\% \pm 2.24\%$ in its best configuration. This enhanced performance of the ST can be attributed to its higher model complexity. Unlike MST, the ST processes the entire multiset as a full list with duplicated elements, thereby incorporating additional nodes and connections to accommodate duplicated elements. This leads to a considerable augmentation in model complexity for the ST, which is potentially tens to hundreds of times greater than that of MST. Such a complexity advantage appears to be particularly beneficial in handling tasks of higher complexity.

It is crucial to highlight that the ST results were not included in Table 2 for real-world datasets. This exclusion stems from the extensive number of points in the PDs of these datasets, surpassing our available computational resources. This limitation emphasizes the necessity for more efficient computational strategies or the development of optimized models to make the ST applicable to larger, real-world datasets. Despite this, we hypothesize that the ST, owing to its higher model complexity, is likely to outperform the MST in most real-world scenarios.

Conversely, the MST offers a more adaptable approach, capable of handling datasets of any size, provided they can be clustered effectively. This flexibility is a notable advantage, especially when dealing with datasets that are too large or complex for the ST to process.

## B  Missing Proofs

### B.1  Proof of the Theorem 5.1

*Proof.* To prove that $A(Q, X)$ is permutation equivariant, we need to show that permuting the rows of $Q$ and $X$ in the same manner results in the same permutation of the rows of $A(Q, X)$.

Let $P$ be an arbitrary permutation matrix. Consider the transformation:

$$A(PQ, PX) = \left( \text{softmax} \left( \frac{PQ(PX)^\top}{\sqrt{d}} \right) + \alpha \frac{(M_{PQ} - \mathbf{1})(M_{PX} - \mathbf{1})^\top}{\|(M_{PQ} - \mathbf{1})(M_{PX} - \mathbf{1})^\top\|_F + \varepsilon} \right) PX.$$

Since $M$ can be seen as a function that operates on the multiset elements, i.e., rows of $Q$ and $X$, the order of the rows in $PQ$ and $PX$ will determine the order of the rows in $M_{PQ}$ and $M_{PX}$, giving:

$$M_{PQ} = PM_Q \quad \text{and} \quad M_{PX} = PM_X.$$

Substituting these into our equation:

$$A(PQ, PX) = \left( \text{softmax} \left( \frac{PQX^\top P^\top}{\sqrt{d}} \right) + \alpha \frac{(PM_Q - \mathbf{1})(PM_X - \mathbf{1})^\top}{\|(PM_Q - \mathbf{1})(PM_X - \mathbf{1})^\top\|_F + \varepsilon} \right) PX.$$

Given that the softmax function is applied element-wise (or row-wise) and retains the order of elements, we can establish:

$$\text{softmax} \left( PHP^\top \right) = P \, \text{softmax} \left( H \right) P^\top.$$

Furthermore, permuting the rows or columns of a matrix only rearranges its elements without changing their values, so the sum of their squared magnitudes, and hence the Frobenius norm, remains unchanged. By equation $\mathbf{1} = P\mathbf{1}$, we have

$$\|(PM_Q - \mathbf{1})(PM_X - \mathbf{1})^\top\|_F = \|(M_Q - \mathbf{1})(M_X - \mathbf{1})^\top\|_F.$$

Substituting these into our equation, we have

$$
\begin{aligned}
A(PQ, PX) &= \left(P\operatorname{softmax}\left(\frac{QX^\top}{\sqrt{d}}\right)P^\top + \alpha\frac{P(M_Q - \mathbf{1})(M_X - \mathbf{1})^\top P^\top}{\|(M_Q - \mathbf{1})(M_X - \mathbf{1})^\top\|_F + \varepsilon}\right)PX \\
&= P\left(\operatorname{softmax}\left(\frac{QX^\top}{\sqrt{d}}\right) + \alpha\frac{(M_Q - \mathbf{1})(M_X - \mathbf{1})^\top}{\|(M_Q - \mathbf{1})(M_X - \mathbf{1})^\top\|_F + \varepsilon}\right)P^\top PX \\
&= P\left(\operatorname{softmax}\left(\frac{QX^\top}{\sqrt{d}}\right) + \alpha\frac{(M_Q - \mathbf{1})(M_X - \mathbf{1})^\top}{\|(M_Q - \mathbf{1})(M_X - \mathbf{1})^\top\|_F + \varepsilon}\right)X \\
&= PA(Q, X).
\end{aligned}
$$

Therefore, $A(PQ, PX) = PA(Q, X)$, which proves that $A(Q, X)$ is permutation equivariant. $\qquad\square$

## B.2 Proof of the Theorem 5.2

*Proof.* As seen in the proof to Theorem 5.1, we have

$$
\begin{aligned}
A_Q(PX) &= \left(\operatorname{softmax}\left(\frac{Q(PX)^\top}{\sqrt{d}}\right) + \frac{M_\alpha(M_{PX} - \mathbf{1})^\top}{\|M_\alpha(M_{PX} - \mathbf{1})^\top\|_F + \varepsilon}\right)PX \\
&= \left(\operatorname{softmax}\left(\frac{QX^\top P^\top}{\sqrt{d}}\right) + \frac{M_\alpha(PM_X - \mathbf{1})^\top}{\|M_\alpha(PM_X - \mathbf{1})^\top\|_F + \varepsilon}\right)PX \\
&= \left(\operatorname{softmax}\left(\frac{QX^\top}{\sqrt{d}}\right)P^\top + \frac{M_\alpha(M_X - \mathbf{1})^\top P^\top}{\|M_\alpha(M_X - \mathbf{1})^\top\|_F + \varepsilon}\right)PX \\
&= \left(\operatorname{softmax}\left(\frac{QX^\top}{\sqrt{d}}\right) + \frac{M_\alpha(M_X - \mathbf{1})^\top}{\|M_\alpha(M_X - \mathbf{1})^\top\|_F + \varepsilon}\right)P^\top PX \\
&= \left(\operatorname{softmax}\left(\frac{QX^\top}{\sqrt{d}}\right) + \frac{M_\alpha(M_X - \mathbf{1})^\top}{\|M_\alpha(M_X - \mathbf{1})^\top\|_F + \varepsilon}\right)X \\
&= A_Q(X)
\end{aligned}
$$

Therefore, $A_Q(PX) = A_Q(X)$, which proves that $A_Q(X)$ is permutation invariant. $\qquad\square$

## C   More on Topological Features

### C.1   Ordinary Persistence and Extended Persistence in Graphs

Given a graph $G = (V, E)$, where $V$ represents the vertices and $E$ denotes the non-oriented edges. Given a function $f : V \to \mathbb{R}$ defined on the vertices of $G$, we can construct sublevel graphs $G_\alpha = (V_\alpha, E_\alpha)$ for each $\alpha \in \mathbb{R}$, such that $V_\alpha = \{v \in V : f(v) \leq \alpha\}$ and $E_\alpha = \{(v_1, v_2) \in E : v_1, v_2 \in V_\alpha\}$. As $\alpha$ increases, we shall observe a sequence of these sublevel graphs, which is referred to as the filtration induced by $f$. This filtration commences with an empty graph and culminates in the entirety of graph $G$. A key aspect of persistence is its ability to track the emergence and dissolution of topological features, such as connected components and loops, throughout this filtration. For example, the birth time, denoted as $\alpha_b$, marks the value at which a new connected component appears in $G_{\alpha_b}$. This component will eventually amalgamate with another at a subsequent value $\alpha_d \geq \alpha_b$, termed the death time. The lifespan of this component is captured by the interval $[\alpha_b, \alpha_d]$. In a similar vein, the birth and death times of loops in specific sublevel graphs are recorded. The aggregation of these intervals forms what is known as the barcode or ordinary PD of $(G, f)$, which can be pictorially represented as a multiset in $\mathbb{R}^2$.

Beyond the conventional scope of persistence, which primarily focuses on sublevel graphs, the concept of extended persistence introduces an additional dimension by considering superlevel graphs. Specifically, for a given $\alpha \in \mathbb{R}$, the superlevel graph $G^\alpha = (V^\alpha, E^\alpha)$ is defined such that $V^\alpha = \{v \in V : f(v) \geq \alpha\}$ and $E^\alpha = \{(v_1, v_2) \in E : v_1, v_2 \in V^\alpha\}$. As $\alpha$ decreases, these superlevel graphs offer an alternative perspective, akin to viewing the same graph from a different direction when juxtaposed with sublevel graphs. This dual perspective ensures a more holistic capture of the topological intricacies inherent in a graph.

Extended PDs, based on the juxtaposition of birth and death points, can be categorized into distinct types. As shown in the main paper, we represent these diagrams as multisets in $\mathbb{R}^2$, denoted as $X_1, X_2, \ldots$, where the subscripted indices signify the different types of PDs. While the above exposition offers a high-level introduction for clarity, readers seeking a deeper understanding are directed to seminal works in the field. Cohen-Steiner et al. (2009) pioneered the concept of extended persistence with rigorous algebraic formulations. Dey & Wang (2022) drew parallels between these types and those found in Zigzag persistence (Carlsson & De Silva, 2010). Carrière et al. (2020) provided insights into these types by interpreting graphs as 1-simplices. For readers with an inclination towards the theoretical underpinnings of persistent homology, we further recommend Edelsbrunner & Harer (2022); Oudot (2017) for an in-depth exploration.

## C.2 Spectral Analysis with Heat Kernel Signature

As alluded to in Subsection 6.2.2, we employ the Heat Kernel Signature (HKS) as filtrations to derive the extended PDs used in our experiments. For the sake of completeness, we provide a brief introduction to HKS here. For an in-depth exploration, we direct readers to seminal works by Carrière et al. (2020), Sun et al. (2009), and Hu et al. (2014).

HKS is a spectral descriptor, originating from the spectral decomposition of graph Laplacians, and has proven to be a powerful tool for graph analysis. Consider a graph $G$ with its vertex set represented as $V = \{v_1, \ldots, v_n\}$. The adjacency matrix of this graph is denoted by $A$. The degree matrix $D$ is a diagonal matrix where each entry $D_{i,i}$ is the sum of the $i$-th row of $A$. The normalized graph Laplacian, $L_n$, is given by the equation

$$L_n = I - D^{-\frac{1}{2}} A D^{-\frac{1}{2}}, \tag{18}$$

where $I$ stands for the identity matrix. This Laplacian possesses an orthonormal basis of eigenfunctions, denoted as $\Psi = \{\psi_1, \ldots, \psi_n\}$, and the associated eigenvalues adhere to the inequality $0 \leq \lambda_1 \leq \ldots \leq \lambda_n \leq 2$. The HKS, parameterized by $t$, is defined as the function $\text{hks}_t$ on the vertices of $G$ by the relation:

$$\text{hks}_t : v \mapsto \sum_{k=1}^{n} \exp(-t\lambda_k)\psi_k(v)^2 \tag{19}$$

It is noteworthy that $\text{hks}_t$ maps vertices to the real line $\mathbb{R}$, thereby inducing a natural filtration for the graph. The parameter $t$ serves as a hyperparameter, the specifics of which are provided in Table 6. The spectral features are a synthesis of the eigenvalues of the normalized graph Laplacian and the deciles of the HKS. Importantly, these features are consistent with those used in the experiment by Carrière et al. (2020).

## D More on The Architecture

This section includes omitted details on the MST.

### D.1 Multihead Attention Mechanism

The multihead attention mechanism, as proposed in (Vaswani et al., 2017), is also a pivotal component of Transformer. The essence of multihead attention lies in its ability to allow different heads to focus on diverse segments of the input data. This is achieved by linearly projecting the $Q, K, V$ vectors into dimensions $d_i, d_i, p_i$ respectively, as outlined by (Vaswani et al., 2017). An added advantage of this projection is the relaxation of the constraint that necessitated $Q$ and $K$ to possess identical dimensions.

To formalize the operation of the multihead attention mechanism, consider the $i$-th head out of a total of $h$ heads. The output for this head is computed as:

$$\text{MultiHead}(Q, K, V) := \text{Concat}\left(\text{h}_1, \ldots, \text{h}_\text{h}\right) W^O, \tag{20}$$

$$\text{h}_i := \text{Att}\left(QW_i^Q, KW_i^K, VW_i^V\right). \tag{21}$$

In the above equations, the projections are characterized by parameter matrices $W_i^Q \in \mathbb{R}^{d_q \times d_i}$, $W_i^K \in \mathbb{R}^{d_k \times d_i}$, $W_i^V \in \mathbb{R}^{d_v \times p_i}$, and $W^O \in \mathbb{R}^{(\Sigma p_i) \times q}$.

This multihead attention mechanism ensures that the model can capture a richer set of relationships and dependencies in the data by allowing each head to focus on different aspects of the input.

For the sake of clarity, the discussions in the paper focuses on the single head attention mechanism. However, it's worth noting that the multihead version can be readily inferred from our descriptions, leveraging the principles delineated in the preceding section.

### D.2   Pre-LN and post-LN MAB

In orchestrating with layer normalization (LN), multiple versions of the multiset attention block (MAB) can be derived. The main paper presents the following formulation:

$$\text{MAB}(Q, X) := \text{LN}(H + \text{FFN}(H)), \tag{22}$$

$$H := \text{LN}(Q + A(Q, X)). \tag{23}$$

In an alternative formulation, we have:

$$\text{MAB}(Q, X) := H + \text{FFN}(\text{LN}(H)), \tag{24}$$

$$H := Q + A\left(\text{LN}(Q), \text{LN}(X)\right). \tag{25}$$

The former is denominated as the post-LN MAB while the latter is termed the pre-LN MAB.

A salient observation is that the application of LN retains the multiplicities $M_X$ in $\text{LN}(X)$. According to the study by Xiong et al. (2020), the Pre-LN Transformers exhibit superior stability relative to the post-LN variants.

For the sake of clarity and to ensure consistency in the ensuing discussions, we will denote both formulations under the generic term MAB due to their functional similarities within the overarching framework.

### D.3   Motivation of Multiset Transformer with Clustering Preprocessing

The primary motivation behind incorporating clustering preprocessing in the MST framework is to balance scalability with the preservation of data details.

A critical insight from the complexity analysis presented in the main paper is the direct correlation between the size of the MST model and the multiset size $n$. Recognizing that neighboring elements in a multiset often hold similar information, clustering these elements before deploying the MST emerges as a strategic move. This preprocessing step stands particularly significant for MST, more so than for the Set Transformer, due to its unique handling of data points.

In MST, the information inherent in the elements is not simply discarded; rather, it undergoes a transformation where it is incorporated into the multiplicities of a representative point. This nuanced approach ensures that crucial data characteristics are retained, albeit in an aggregated form. Consequently, we adopt a preprocessing strategy where multisets are clustered prior to their engagement with the MST. This methodology not only maintains comparable performance levels but also leads to a significant reduction in computational complexity, thereby enhancing the overall efficiency of the model.

# E  Experimential Details

This section provides further details regarding these experiments, including information on datasets and hyperparameters.

## E.1  Datasets

Table 5 provides a summary of the information for each dataset.

Table 5: Dataset descriptions. Here, $\beta_0$ and $\beta_1$ represent the 0th and 1st Betti numbers, indicating the number of connected components and cycles in a graph, respectively. Specifically, an average $\beta_0 = 1.0$ denotes that all graphs in the dataset are connected. In these cases, $\beta_1 = \#\{\text{edges}\} - \#\{\text{nodes}\}$. Source: Carrière et al. (2020).

| Dataset | Nb graphs | Nb classes | Av. nodes | Av. Edges | Av. $\beta_0$ | Av. $\beta_1$ |
|---|---|---|---|---|---|---|
| MUTAG | 188 | 2 | 17.93 | 19.79 | 1.0 | 2.86 |
| COX2 | 467 | 2 | 41.22 | 43.45 | 1.0 | 3.22 |
| DHFR | 756 | 2 | 42.43 | 44.54 | 1.0 | 3.12 |
| NCI1 | 4,110 | 2 | 29.87 | 32.30 | 1.19 | 3.62 |
| NCI109 | 4,127 | 2 | 29.68 | 32.13 | 1.20 | 3.64 |
| PROTEIN | 1,113 | 2 | 39.06 | 72.82 | 1.08 | 34.84 |
| IMDB-B | 1,000 | 2 | 19.77 | 96.53 | 1.0 | 77.76 |
| IMDB-M | 1,500 | 3 | 13.00 | 65.94 | 1.0 | 53.93 |
| COLLAB | 5,000 | 3 | 74.5 | 2457.5 | 1.0 | 2383.7 |

## E.2  Devices

We utilized both an RTX 4090 (24 GB) and an RTX A6000 (48 GB) GPU for our experiments.

## E.3  Hyperparameters

The experiments in this study consistently employed specific hyperparameters to guarantee replicability. Details of these hyperparameters can be found in Table 6. In the table, $\text{hks}_t$ denotes the functions used to create filtrations. These filtrations are essential as they are later used to derive PDs from graphs. Additionally, an initial random seed of 42 was used throughout the experiments.

Table 6: Hyperparameters for various datasets. H: Number of attention heads in MABs; E: Equivariant layers; E. Q: Queries in IMABs for equivariant layers; I. Q: Queries in $\text{MAB}_Q$ for invariant layer; Pre-LN: MAB version (True/False for using pre-layer normalization); Eps: Maximum sample distance in DBSCAN neighborhood; Hidden: Hidden units; LR: Learning rate; Epochs: Training epochs; Batch: Training batch size.

| Dataset | HKS | H | E | E. Q | I. Q | Pre-LN | Eps | Hidden | LR | Epochs | Batch |
|---|---|---|---|---|---|---|---|---|---|---|---|
| SYNTHETIC | - | 2 | 2 | 1 | 4 | False | - | 64 | 0.01 | 100 | 128 |
| MUTAG | $\text{hks}_{10}$ | 2 | 2 | 2 | 4 | True | 0.5 | 64 | 0.01 | 150 | 128 |
| COX2 | $\text{hks}_{0.1}, \text{hks}_{10}$ | 2 | 2 | 2 | 8 | False | 0.5 | 64 | 0.02 | 200 | 128 |
| DHFR | $\text{hks}_{0.1}, \text{hks}_{10}$ | 2 | 2 | 4 | 8 | True | 0.5 | 64 | 0.01 | 200 | 128 |
| NCI1 | $\text{hks}_{0.1}, \text{hks}_{10}$ | 2 | 2 | 8 | 16 | True | 0.1 | 256 | 0.06 | 300 | 128 |
| NCI109 | $\text{hks}_{0.1}, \text{hks}_{10}$ | 2 | 2 | 8 | 16 | True | 0.1 | 64 | 0.1 | 100 | 128 |
| PROTEIN | $\text{hks}_{10}$ | 2 | 2 | 2 | 8 | True | 0.01 | 64 | 0.01 | 200 | 128 |
| IMDB-B | $\text{hks}_{0.1}, \text{hks}_{10}$ | 2 | 2 | 2 | 8 | False | 0.04 | 64 | 0.01 | 100 | 128 |
| IMDB-M | $\text{hks}_{0.1}, \text{hks}_{10}$ | 2 | 2 | 2 | 8 | True | 0.04 | 64 | 0.01 | 100 | 128 |
| COLLAB | $\text{hks}_{0.1}, \text{hks}_{10}$ | 2 | 2 | 1 | 8 | True | 0.01 | 64 | 0.01 | 100 | 128 |

