# OpenReview forum: "Multiset Transformer: Advancing Representation Learning in Persistence Diagrams"
_TMLR — Rejected by TMLR_

### Review · Reviewer_BEXP · 2024-06-10

**Summary Of Contributions:**

The paper describes a tweaked MHA that incorporates multiplicity information into the Attention mechanism by effectively performing a 2nd QK^T operation only on the multiplicity map (using the normalized dot product representation of cosine similarity kernel). The method is evaluated on a toy dataset and the datasets from https://proceedings.mlr.press/v108/carriere20a.html, which is used as a baseine. The authors claim improved performance compared to https://proceedings.mlr.press/v108/carriere20a.html and simple set transformer baseline.

**Audience:**

No

**Broader Impact Concerns:**

Not required

**Claims And Evidence:**

No

**Requested Changes:**

## Critical

1. perform the statistical tests
2. add at least one sota linear complexity baseline (e.g. exphormer, another linear attention)
3. explain mismatch with carriere et al 2020
4. add missing citation

## Suggested

Tighten the papers language, replace re-exposition of MHA with appendix C (less people know TDA than MHA), add more experiments/illustrations (what about non-graph data that might exhibit topological features?)

# Nice to have

- clarify multiple seeds

**Strengths And Weaknesses:**

Going from very weak (---) to very strong (+++):

\- intro could use a one sentence description of PD, or a reference to the appendix C
\- a lot of the paper reads padded and wordy. it's 2024, we do not need more than a page rederiving MHA and the MAB block or the concept of chaining equivariant and invariant layers, likewise the complexity analysis is trivial.
\--- The space could be used to show table 4 in the main body, as it is it looks like a missing baseline in table1, compare against standard transformer with similar parameter count. Also, "Ratio" is redundantly defined, is $$\vert X \vert$$ cardinality or absolute value?
\--- ST vs. MST looks awfully close, please perform a suitable statistical test (e.g. https://en.wikipedia.org/wiki/Welch%27s_t-test ) with multiple comparison correction to ensure results are statistically significant (you might need more seeds)
\-- table 2 of https://proceedings.mlr.press/v108/carriere20a/carriere20a.pdf indicates perslay achieves 74.8 on proteins not 69.7 or 72.2. this without explanation plus missing std casts doubts on the implementation, was the original papers code used?
\-- no baseline to vanilla GAT, exphormer or the other sota models, without justification (I see computational constraints in the appendix for ST, which is valid, but the aforementioned are all linear complexity)
\- are the 5 runs across multuple seeds?
\- missing discussion of https://arxiv.org/abs/2111.12193 as a multiset equivariant network

---

> ### Author Response · Authors · 2024-09-16
>
> We appreciate the reviewer's thoughtful feedback. We have carefully considered the reviewer's suggestions and would like to address them as follows:
>
> > --- The space could be used to show table 4 in the main body, as it is it looks like a missing baseline in table 1, compare against standard transformer with similar parameter count. Also, "Ratio" is redundantly defined, is cardinality or absolute value?
>
> **Placement of Table 4:**
> We initially included the full Table 4 in the main body of our draft. However, we later moved it to the appendix for the following reasons:
> - The comparison between MST and ST is partial, and including it in the main body might raise questions about the absence of ST in the main Table 2.
> - We aimed to present a clear focus to readers: MST as a NN-based vectorization method offers unique advantages over previous approaches (e.g., PERSLAY).
>
> We condensed Table 1 and placed the full table in the appendix to maintain this focus. However, we are open to reincorporating it into the main body if deemed necessary.
>
> **Clarification of "Ratio":**
> We acknowledge that the term "Ratio" needed further clarification. In the paragraph below Table 1, we have refined the introduction of this concept. Specifically, we have adjusted it
>
> from
>
> *In the table, we use the data ratio $\frac{|X|}{\|M_X\|_1}$, which represents the ratio of the count of unique items to the sum of their multiplicities, to highlight the size difference between set and multiset representations.*
>
> to
>
> *In the table, the data ratio is defined as $\frac{|X|}{\|M_X\|_1}$, where $|X|$ is the cardinality of the base set $X$. It represents the ratio of the count of unique items to the sum of their multiplicities, highlighting the size difference between set and multiset representations.*
>
> > --- ST vs. MST looks awfully close, please perform a suitable statistical test (e.g. https://en.wikipedia.org/wiki/Welch%27s_t-test ) with multiple comparison correction to ensure results are statistically significant (you might need more seeds)
>
> You are correct that the results of ST and MST appear close in some cases. We have conducted a statistical analysis using Welch's t-test with multiple comparison correction to address this concern. Here are the results:
>
> | # Classes | MST (inv.)        | ST                | p-value |
> |-----------|-------------------|-------------------|---------|
> | 2         | **100.00** ± 0.00 | **100.00** ± 0.00 | -       |
> | 3         | **99.88** ± 0.15  | 99.74 ± 0.15      | 0.18    |
> | 5         | **88.86** ± 2.07  | 85.82 ± 1.73      | 0.036   |
> | 11        | 41.14 ± 2.24      | **42.02** ± 1.85  | 0.52    |
>
> The statistical analysis reveals that:
>
> 1. For 2 classes, both methods achieve perfect performance.
> 2. For 3 and 5 classes, MST slightly outperforms ST, though the difference is only statistically significant (p < 0.05) for 5 classes.
> 3. For 11 classes, ST marginally outperforms MST, but the difference is not statistically significant.
>
> These results indicate that the performance of MST and ST is indeed close, and we cannot draw a unanimous conclusion regarding their comparative effectiveness across all scenarios.
>
> It's worth noting that MST can be viewed as a generalization of ST, as MST degenerates to ST when the input multiset is a set. This perspective helps explain why their performances are similar when inputs are sets. However, when inputs are multisets, ST models tend to be larger due to item duplication, potentially offering more expressive power. MST provides a way to compensate for this without increasing model size.
>
> We acknowledge that the real-world performance dynamics between MST and ST are complex and may depend on various factors.
>
> Continue in the following comment...

---

> ### Author Response · Authors · 2024-09-16
>
> > -- table 2 of https://proceedings.mlr.press/v108/carriere20a/carriere20a.pdf indicates perslay achieves 74.8 on proteins not 69.7 or 72.2. this without explanation plus missing std casts doubts on the implementation, was the original papers code used?
>
> We appreciate the reviewer's thorough comparison and the opportunity to clarify our results.
>
> For a fair comparison, we used the results from the "PD alone" column in Table 7 of the supplementary material (https://proceedings.mlr.press/v108/carriere20a/carriere20a-supp.pdf), which align with the values presented in the Table 2. This way, we can guarantee that all the data input to the MST are exactly the same as PERSLAY, without the effect of optional feature input.
>
> Thank you for bringing this to our attention. We have added a description of the exact source of PERSLAY in Table 2 to improve clarity. More specifically, we added
>
> *All the results of PERSLAY are copied from \citet[Table 7]{carriere2020perslay}.*
>
> to the table caption.
>
> > -- no baseline to vanilla GAT, exphormer or the other sota models, without justification (I see computational constraints in the appendix for ST, which is valid, but the aforementioned are all linear complexity)
>
> We acknowledge that comprehensive comparisons are valuable for readers and the research community.
> The paper doesn't include these comparisons at this moment primarily due to the following reasons:
>
> 1. **Focus on MST contribution:** The main argument of the paper is that MST provides an improved NN-based vectorization method for persistence diagrams. Comparisons with GAT, Exphormer, or other state-of-the-art GNN-based models, while valuable, would not directly support this central claim.
>
> 2. **Scope and time constraints:** Implementing and fairly comparing with these models would require significant architectural modifications and potentially shift the focus towards overall model performance rather than the specific contribution of MST. Such extensive work is beyond the scope of what could be accomplished during the rebuttal period.
>
> We acknowledge that these additional comparisons could provide broader context and will consider including them in future work. We appreciate the reviewer's understanding and would like to hear more suggestions on how to best demonstrate the value of the MST approach within the current framework of the study.
>
> > \- are the 5 runs across multuple seeds?
>
> We confirm that the 5 runs were conducted across multiple seeds. The experiment generates a unique random seed for each run in a sequential loop, ensuring different seeds for each of the 5 runs.
>
> > \- missing discussion of https://arxiv.org/abs/2111.12193 as a multiset equivariant network
>
> We added a discussion when we introduce permutation equivariance. Specifically, we added
>
> *While \citet{zhang2021multiset} introduced the above permutation equivariance as set equivariance and further proposed multiset-equivariance as its relaxation, our architecture distinctly separates the base set from its multiplicities. This separation results in set like inputs, making set equivariance the appropriate property for our approach.*
>
> We appreciate the reviewer's attention to detail, which has helped us provide a more nuanced analysis of our results.

---

### Review · Reviewer_P13E · 2024-06-14

**Summary Of Contributions:**

This paper proposes the Multiset Transformer designed for multisets as inputs with extensive numerical verifications.

**Audience:**

Yes

**Claims And Evidence:**

Yes

**Requested Changes:**

See weakness

**Strengths And Weaknesses:**

Strengths:

1. The paper is well-written and provides robust numerical support.

2. The connection with the conventional Transformer is clearly and effectively presented.

Weakness:

1. It is unclear whether this new model architecture can be implemented in parallel computing.

2. The authors should conduct and report a comparison with the vanilla Transformer. Additionally, it is important to present the final values of the learned parameter $M_X$ to demonstrate whether the observed improvements are indeed attributable to this learnable parameter.

---

> ### Author Response · Authors · 2024-09-16
>
> We appreciate the reviewer's insightful comments and suggestions. Here are the replies.
>
> > 1. It is unclear whether this new model architecture can be implemented in parallel computing.
>
> We would like to clarify that the architecture maintains a level of parallelism comparable to the Set Transformer. The computations within individual neural network modules (e.g., FFN, LayerNorm, or the Multiset Attention) can be efficiently parallelized. However, due to data flow dependencies, parallelization between modules is limited, which is a common constraint in neural network architectures.
> the code is implemented in PyTorch, leveraging its built-in parallelization capabilities to the fullest extent possible.
>
> > 2. The authors should conduct and report a comparison with the vanilla Transformer. Additionally, it is important to present the final values of the learned parameter to demonstrate whether the observed improvements are indeed attributable to this learnable parameter.
>
> **Comparison with vanilla Transformer:**
> While we acknowledge the value of such a comparison, it is currently beyond our computational resources due to the complexity hierarchy of these models:
> $\text{Transformer} \ge \text{Set Transformer} \ge \text{Multiset Transformer}.$
>
> However, we have conducted a partial comparison between the Multiset Transformer (MST) and the Set Transformer (ST) on synthetic datasets, where we could control data size. These results are presented in Table 4.
>
> **Learned parameter effectiveness:**
> The effectiveness of the learnable parameter $\alpha$ or the parameters in the bias term can be observed in the ablation study provided in subsection 6.2.5, particularly in Table 3. The comparison between the "MST" and "MST (w/o mult.)" columns demonstrates the impact of this parameter.
>
> "MST (w/o mult.)" refers to MST without multiplicities input, equivalent to setting the bias term in Equation (6) to zero (or $\alpha = 0$) and making the multiset biased attention void. The performance difference between these two columns illustrates that the observed improvements can indeed be attributed to this learnable parameters.
>
> We appreciate the reviewer's insights and hope this clarification addresses their concerns.

---

### Review · Reviewer_c3HW · 2024-09-04

**Summary Of Contributions:**

This paper introduces a novel Multiset Transformer (MST) and its subsequent application in PD representation learning which utilizes the multiplicities in multiset inputs while significantly reducing spatial and computational complexity compared to the Set Transformer. Experimental results show that MST inherently offers an approximation by clustering the multiset prior to processing.

**Audience:**

Yes

**Broader Impact Concerns:**

Not applicable.

**Claims And Evidence:**

Yes

**Requested Changes:**

Please check comments of Weaknesses.

**Strengths And Weaknesses:**

Strengths:

+ The manuscript is easy to read and well-written.
+ Multiset transformer architecture is interesting and corresponding idea has been explained well.
+ Computational complexity is provided.

Weaknesses:

- The authors can consider using more state-of-the-art models for comparison instead of only using PERSLAY.
- It would be good if the authors apply the proposed MST model to large-scale datasets such as OGB.
- Could the authors comment on how to select optimal hyperparameters in the experiments?

---

> ### Author Response · Authors · 2024-09-16
>
> We appreciate the reviewer's insightful comments and suggestions. The following are the replies to the concerns.
>
> > The authors can consider using more state-of-the-art models for comparison instead of only using PERSLAY.
>
> We acknowledge that the experimental section may appear limited in its scope. This is primarily due to the interdisciplinary nature of the topic - vectorizing persistent diagrams via neural networks - which has been explored in only a couple of papers.
>
> Among these, PERSLAY stands out as the state-of-the-art model. It not only provides a benchmark for comparison but also supplies the **exact persistent diagram data** that can be directly inputted into the proposed MST model. This allows for a precise and fair comparison between PERSLAY and MST, which we believe is a significant advantage.
>
> > It would be good if the authors apply the proposed MST model to large-scale datasets such as OGB.
>
> Regarding the application of the MST model to large-scale datasets such as OGB, we agree that this would be a valuable addition. We will consider this in the future work to further validate the effectiveness of our model, given the time constraints.
>
> > Could the authors comment on how to select optimal hyperparameters in the experiments?
>
> The approach to hyperparameter tuning follows a systematic process. Initially, we define the search space based on insights from previous experiments, specifically referencing methodologies from PERSLAY and Set Transformer. We primarily employ grid search to explore this space. Additionally, we incorporate random search and manual tuning to further refine the hyperparameter selection.
>
> These combined methods enable us to identify hyperparameters that are close to optimal, as detailed in Table 6 for replicability purposes. However, we acknowledge that we cannot guarantee these hyperparameters are the absolute optimal. Factors such as the number of layers can significantly alter the neural network's size and complexity. Consequently, we constrained the search space to ensure that our computational resources could efficiently train the models.
>
> In summary, while the hyperparameters we selected are not guaranteed to be optimal, they are chosen to be as close to optimal as possible within the limitations of our available resources.
>
> We appreciate the reviewer's insights and hope this clarification addresses the concerns.

---

### Decision · Action_Editor_XECA · 2024-11-15

**Recommendation:** Reject

**Comment:**

For this manuscript, two of the three reviewers lean towards acceptance and one towards rejection.

Some aspects of the paper were appreciated by reviewers:

+ The architecture was considered interesting [c3HW] and the connection with the Transformer was considered clearly presented [P13E]

Some aspects had a mixed reception:

+ The manuscript was considered well written [c3HW,P13E] and easy to read [c3HW] by two reviewers, however one reviewer considered the writing 'padded' and 'wordy' [BEXP]
+ Providing the computational complexity was appreciated by one reviewer [c3HW], but the complexity analysis was considered trivial by another reviewer [BEXP]

Several downsides were identified, many of which persisted despite t the author response:

- Comparisons to more state of the art models beyond PERSLAY were desired [c3HW,BEXP]
- Comparisons on large scale data sets were desired [c3HW]
- Discussion of selection of the hyperparameters in the experiments was desired [c3HW]
- Discussion of possibility of parallel implementation was desired [P13E]
- Comparison with the vanilla Transformer was desired [P13E,BEXP]
- Statistical significance testing of differences was desired [BEXP]
- Possible disrepancies between results reported here and elsewhere were noted and discussion of them was desired [BEXP]
- Presentation and analysis of the learned M_x values was desired [P13E]

Although one reviewer felt the author rebuttal strengthened the manuscript [BEXP] the reviewer still felt the concerns of missing significance testing and lack of additional baselines remained.

Ultimately, there are strong enough concerns remaining that the manuscript cannot be considered acceptable in its current form or with a minor revision. A sufficient revision might be able to address the remaining concerns.

**Audience:**

The overall idea of the new architecture seems to be appreciated by reviewers, and could find an audience in TMLR if the manuscript quality was otherwise at a sufficient level; see Commends for details.

**Claims And Evidence:**

There are several concerns about the convincingness of the evidence (see Comments for details).

**Resubmission Of Major Revision:**

The authors may consider submitting a major revision at a later time.